# Sickle Cell Disease in the Islands of Zanzibar: Patients’ Characteristics, Management, and Clinical Outcomes

**DOI:** 10.3390/genes16010047

**Published:** 2025-01-02

**Authors:** Ahlam Amour, Fadya Hashim, Fat-hiya Said, Daniel Joshua, Daniel Kandonga, Michael Msangawale, Agnes Jonathan, Benson Kidenya, Paschal Ruggajo, Marijani Msafiri, Emmanuel Balandya, Muhiddin Mahmoud

**Affiliations:** 1Mnazi Mmoja Hospital (MMH), Kaunda Road, Vuga Street, Zanzibar 71102, Tanzania; fadyashineni@hotmail.com (F.H.); saidfathiya69@gmail.com (F.-h.S.); msafirimarijani@yahoo.com (M.M.); muhiddinabdi@gmail.com (M.M.); 2Sickle Cell Program, Muhimbili University of Health and Allied Sciences, Upanga, Dar es Salaam 11103, Tanzania; dkandonga@blood.ac.tz (D.K.); mmsangawale@blood.ac.tz (M.M.); ajonathan@blood.ac.tz (A.J.); benkidenya@yahoo.com (B.K.); prugajo@yahoo.com (P.R.); ebalandya@muhas.ac.tz (E.B.); 3Zanzibar Heath Research Institute (ZAHRI), 87 Tunguu Kibaoni Road, South Unguja, Zanzibar 72214, Tanzania; danieljoshua36@yahoo.com; 4Department of Biochemistry and Molecular Biology, Catholic University of Health and Allied Sciences (CUHAS), Bugando, Mwanza 33830, Tanzania; 5Medical College, The Aga Khan University, Upanga, Dar es Salaam 11102, Tanzania

**Keywords:** sickle cell disease, Zanzibar, clinical characteristics, management, outcomes

## Abstract

Background: This study aimed to describe Sickle Cell Disease (SCD) phenotypes, sociodemographic characteristics, healthcare, and clinical outcomes of patients with SCD attending Mnazi Mmoja Hospital (MMH) in Zanzibar. Methods: Individuals who visited MMH between September 2021 and December 2022 and were known or suspected to have SCD were enrolled in the clinic. Sociodemographic characteristics and clinical features were documented, and laboratory tests were performed. A two-sample test of proportions was used to ascertain the significance of differences in the distribution of clinical outcomes between the follow-up visits. Results: A total of 724 patients with SCD were enrolled: 367 (50.7%) were male, and 357 (49.3%) were female. Most patients—713 (98.5%) in total—were homozygous (Hb SS), 9 (1.2%) had the Hb SC phenotype, and 2 (0.3%) had HbS β+ thalassemia. The majority of patients were aged 13 years and below—520 (71.8%) in total—and most did not have health insurance—582 (80.4%) in total. While all patients received folic acid, only a quarter received pneumococcal prophylaxis and hydroxyurea. Attendance at the third visit was associated with a reduced frequency of self-reported episodes of pain (24 patients [4.3%] vs. 11 patients [1.9%]). Conclusion: The population of patients with SCD in Zanzibar mostly comprised children who were Hb SS. Basic care services are still suboptimal, although they are associated with better outcomes when present. Thorough evaluation of SCD prevalence in Zanzibar through newborn screening programs is warranted.

## 1. Introduction

Sickle cell disease (SCD) is an inherited genetic disorder that affects hemoglobin molecules in red blood cells. It is caused by a mutation in the hemoglobin subunit β (HBB) gene, leading to the production of abnormal hemoglobin, which causes red blood cells to become rigid and crescent-shaped under low oxygen tension, leading to the blockage of small blood vessels [1]. The blockage of small blood vessels can cause a variety of complications such as severe pain, organ damage, and increased risk of infection [2]. These symptoms vary greatly, ranging from mild to severe [3].

The World Health Organization (WHO) specifically focuses on SCD and works to improve the lives of people living in these conditions [4]. The major focus of the WHO includes efforts to improve access to diagnosis, treatment, and care for people living with SCD, particularly in low- and middle-income countries where access to medical care may be limited [5]. The WHO also aims to raise awareness of the condition and promote the research and development of new treatments.

According to the WHO, SCD affects millions of people worldwide, with the highest prevalence occurring in sub-Saharan Africa (SSA). The number of people living with SCD globally increased by 41.4% (from 5.46 million in 2000 to 7.74 million in 2021), with babies born with SCD having increased by 13.7% to 515,000, while in SSA, over 300,000 babies are born with SCD annually [6]. The highest burden of SCD in SSA is found in countries such as Nigeria, Angola, Tanzania, and the DRC [7]. In developed countries, SCD is most commonly found in individuals of African descent; however, it also affects people of Hispanic, Mediterranean, and Middle Eastern descent [8]. In the United States of America, it is estimated that around 100,000 individuals have SCD [8,9]. The incidence of SCD is lower in developed countries than in developing countries; however, many individuals living with SCD in developed countries face challenges similar to those in developing countries, such as a lack of access to proper medical care and treatment, limited awareness of the condition, and cultural stigmatization of the disease [10].

SCD is a serious and debilitating condition requiring lifelong management and care. Early diagnosis and treatment can help to prevent complications and improve patient outcomes [11]. Access to proper medical care, adequate pain management, and early detection and treatment of infections can save lives and help improve the quality of life of people with SCD. Support groups and community-based programs can also provide emotional and psychological support to individuals with SCD and their families [12,13]. However, the lack of access to proper medical care and treatment in some areas of the world, limited awareness of the condition, and cultural stigmatization of the disease are some of the challenges faced by people living with SCD [13,14].

Access to proper medical care and treatment for SCD is limited in developing countries, leading to poor health outcomes for those living with the disease. Many sufferers do not have access to proper diagnosis and treatment and often rely on traditional healers for care [10]. Additionally, the cost of care for SCD can be high, which can make it difficult for low-income families to afford necessary treatment [15]. In Tanzania, limited knowledge of SCD among healthcare workers and the limited availability of essential drugs and equipment are some of the challenges reported to hinder proper care for patients with SCD [16,17,18]. Improper care causes many individuals living with SCD in developing countries like Tanzania to struggle with chronic pain, reduced physical activity, and a lower quality of life [19,20].

Despite these challenges, efforts are being made to improve SCD management and care in developing countries. These include the development of guidelines for the management of SCD [21,22], awareness campaigns [23], providing training and education for healthcare workers [24], facilitating access to essential medicines and technologies, supporting newborn screening programs [25,26], and conducting research to improve the understanding of the disease. Furthermore, the implementation of newborn screening (NBS) programs, availability of hydroxyurea, and increasing the number of trained healthcare providers can help improve the overall outcomes of people living with SCD [25,27,28].

In Tanzania, the estimated birth prevalence of SCD is between 0.8% and 1.4%, with a sickle cell gene carrier rate (sickle cell trait) of 13–20% [29,30]. Zanzibar, an archipelago off the coast of Tanzania, is thought to have a high prevalence of SCD due to the island’s history of malaria and migration, which results in a high frequency of sickle cell traits among the population. However, research on SCD in Zanzibar is limited. Available studies have focused on the carrier rate of HBB gene mutations that cause SCD [31] and on understanding caregivers’ knowledge of SCD and home-based care practices [32]. This article aims to highlight the SCD phenotypes, sociodemographic characteristics, healthcare, and clinical outcomes of patients with SCD attending the Mnazi Mmoja Hospital (MMH), which is a tertiary referral hospital in Zanzibar.

## 2. Materials and Methods

### 2.1. Study Design

This was a prospective cohort study involving patients with SCD who attended the Mnazi Mmoja Hospital (MMH) outpatient clinic through the Sickle Pan-African Research Consortium (SPARCO)-Tanzania project. Patient data were prospectively collected from September 2021 to December 2022 and recorded in a REDCap-based database.

### 2.2. Study Setting

This study was conducted at MMH in Zanzibar, Tanzania. MMH is a public tertiary referral hospital located on Unguja Island, the largest of the two islands in Zanzibar, serving the entire 1.8 million population of Zanzibar. The hospital offers a range of medical services, such as diagnosis and treatment of various health conditions, including SCD. MMH has a bed capacity of 760 beds and is unique in that it is the sole healthcare facility in Zanzibar that provides comprehensive SCD care. The hospitals conduct 25 outpatient clinics for SCD, including dedicated pediatric and adult clinics. SCD clinics are conducted on Wednesdays (for children aged 0–12 years) and Fridays (for adolescents aged 13 years and adults). Routine services available at outpatient SCD clinics include health education, clinical assessment, laboratory assessment (hematology and biochemistry), imaging, and the provision of essential medications (hydroxyurea, folic acid, antimalarial prophylaxis, and penicillin V prophylaxis for children up to 5 years of age). Since 2012, the Pneumococcal Conjugate Vaccine (PCV)-13 has been given free of charge to all infants in Tanzania as part of the Expanded Program on Immunization (EPI), regardless of SCD status. Booster pneumococcal vaccines for individuals with SCD are administered using Pneumococcal Polysaccharide Vaccine (PPSV)-23, which is purchased out of pocket. SCD tests available at the hospital include point-of-care tests such as Sickle SCAN (BioMedomics Inc, Morrisville, North Carolina, USA) and Hb electrophoresis. The hospital’s department of pediatrics also has dedicated inpatient services for children with SCD. Furthermore, the hospital has specialized inpatient and outpatient services in hematology, internal medicine (including a dialysis facility), obstetrics and gynecology, and general surgery, where patients with SCD can also attend. Individuals with health insurance access most of their services through insurance plans. The government bears the costs of medical care for those without health insurance. In the absence of certain services or medications through government plans, patients pay for out-of-pocket services or medications.

### 2.3. Study Variables

In this study, data on several variables were collected and analyzed to understand the status of SCD in Zanzibar. Primary variables included SCD phenotypes and patient demographic information, such as age and sex. Other variables included healthcare and clinical information, such as ownership of health insurance, medications used, severity of anemia (Hb < 5 gL/dL), frequency of pain episodes, hospitalization, and blood transfusion.

### 2.4. Data Collection

The data for this study were collected from the medical records of patients with SCD at MMH in the SPARCO-Tanzania REDCap database. Demographic and clinical data (self-reported and laboratory results) were collected prospectively during routine clinic visits and recorded in a standardized format. SCD status was determined using the Sickle SCAN and Hb electrophoresis. All data were verified for accuracy and completeness before the analysis. Trained research staff who followed strict protocols collected the data to ensure that they were valid and reliable.

### 2.5. Statistical Analysis

The collected data were analyzed using descriptive statistics to summarize the demographic, healthcare, and clinical outcomes in frequencies and corresponding percentages, and they are presented in tables. Individuals presenting with a hemoglobin level of <5 g/dL were considered to have severe anemia. We also used a two-sample test of proportions to ascertain the significance of the differences in the distribution of various clinical outcomes (severity of anemia and frequency of pain episodes, hospital admissions, and blood transfusions) between the number of follow-up visits. Data were analyzed using STATA version 15. Statistical significance was set to 0.05.

### 2.6. Ethical Considerations

This study was approved by the Muhimbili University of Health and Allied Sciences (MUHAS) Research Ethics Committee (MUHAS-REC-12-2020-453) and National Health Research Ethics Committee (NatHREC-NIMR/HQ/R.8a/Vd. Lx/3818). Permission to conduct this study was obtained from the Management of the MMH. Written informed consent was obtained from all the patients. Confidentiality of the patients’ information was maintained throughout the study, and all data were handled in accordance with the ethical and legal requirements.

## 3. Results

### 3.1. Sociodemographic Characteristics of Patients with SCD at MMH

In total, 724 patients were recruited during the study period. Of these, 367 (50.7%) were male, and 357 (49.3%) were female. Most patients were aged 0–13 years, 520 (71.8%), and 123 (17.0%) were aged ≥ 18 years. The age group of 14–17 years had the lowest number of patients at 81 (11.2%). Seven hundred and thirteen patients (98.5%) had a homozygous Hb SS phenotype, nine (1.2%) had a homozygous Hb SC phenotype, and only two (0.3%) had Hb S β+thalassemia. The majority of the SCD patients 582 (80.4%) did not have health insurance (Table 1).

### 3.2. Clinical Management of Patients with SCD at MMH

All patients in this study received folic acid, whereas only 192 (26.6%) were up to date on pneumococcal vaccination. The number of patients receiving hydroxyurea was 191 (26.4%), whereby penicillin V and anti-malarial prophylaxis were prescribed to 188 (25.9%), and 4 (0.5%) patients received both penicillin V and antimalarial prophylaxis (Table 2).

### 3.3. Association Between Follow-Up Visits and Clinical Outcomes Among Patients with SCD at MMH

In this study, we observed a statistically significant difference in the number of patients who reported experiencing pain during the second visit compared to the third visit, whereby 24 patients (4.3%) reported pain during the second visit compared to 11 patients (1.9%) during the third visit (*p* = 0.021). In contrast, hospital admissions were slightly lower at the second visit than at the third visit (27 [4.6%] vs. 32 [5.46%] patients; *p* = 0.042). The proportion of patients with severe anemia at the third visit was slightly lower than at the second visit, although the difference was not statistically significant. Similarly, the proportion of patients who underwent blood transfusions did not differ between the second and third follow-up visits (Table 3).

## 4. Discussion

Understanding the clinical epidemiology of a disease is an important step toward implementing appropriate interventions. Here, our findings suggest a predominant occurrence of the Hb SS phenotype among SCD patients in Zanzibar, most of whom were children. Most patients were unable to receive basic healthcare services such as pneumococcal prophylaxis and hydroxyurea. Nonetheless, attendance at follow-up clinics was associated with fewer episodes of pain crisis, highlighting the potential benefits of improving clinical care services for SCD patients on the islands.

Homozygous SCD (Hb SS) was the most dominant form of SCD in our population, although other phenotypes, including Hb SC and Hb S β+ thalassemia, were also observed. The overwhelming dominance of Hb SS is likely due to the high prevalence of individuals with sickle cell traits in Tanzania, approaching 20% in areas around Lake Victoria [29]. On the other hand, the presence of β thalassemia likely reflects the shared Middle Eastern heritage in the region following its colonial past [33].

Most patients were aged 0–13 years (71.8%). The over-representation of children in our cohort was similar to the observations made in other SCD cohorts in SSA [20,32]. However, it is likely that the proportion of children in our cohort does not necessarily reflect the actual landscape of SCD in Zanzibar for several reasons. First, our cohort was based at Mnazi Mmoja Hospital, which is Zanzibar’s tertiary referral hospital, implying that patients with possibly unique sociodemographic characteristics may have sought treatment at other private or governmental facilities and thus were not included in our study. Second, it is possible that some parents are still reluctant to bring their children to hospitals because of a lack of awareness of SCD and misconceptions about the disease or stigma [34]. As a result, it is possible that we did not record all representative groups of patients with SCD in our study.

Although most patients in our cohort received folic acid supplements, nearly three-quarters did not receive hydroxyurea, penicillin V prophylaxis, or PPSV-23 booster vaccines. Limited access to these important treatments is attributed to multiple barriers. A key challenge includes the high cost of these therapies, which places them out of reach for many patients, as well as a logistical challenge within the health care system that limits a consistent supply chain and distribution [35,36,37]. Furthermore, misconceptions and knowledge gaps among patients, caregivers, and healthcare workers are other possible barriers that hinder the use of these medications [37,38]. Similar studies in Mainland Tanzania and Malawi have documented significant gaps in the uptake of hydroxyurea due to high costs, limited availability, and widespread misconceptions [17]. Addressing these barriers requires not only policy and financial interventions to improve accessibility but also health education in the community to combat myths and misinformation.

Similarly, anti-malarial prophylaxis was prescribed to only 0.6% of the SCD patients. With the islands moving towards Zero Malaria, the prevalence of malaria in Zanzibar is very low (currently below 1%), and this likely accounts for the low utilization of malaria prophylaxis. Efforts are underway through SPARCO-Tanzania and other initiatives to increase the awareness of SCD among patients, caregivers, and the general public, as well as advance the standards of care for SCD through capacity building among healthcare workers. Our study observed a statistically significant reduction in self-reported episodes of pain from the second toward the third visit compared to the previous visit. This indicates favorable outcomes as a result of long-term care retention. In contrast, we observed a slight increase in the proportion of patients reporting hospital admission from the second toward the third visit compared to the prior visit. While on one hand this may suggest unfavorable outcomes over time, it may also indicate heightened awareness among patients and an improvement in health-seeking behavior [38,39]. We also observed a reduction in the proportion of patients with severe anemia over the same period of time, although this did not reach statistical significance. Collectively, our data suggest a general improvement in clinical outcomes among patients with SCD following long-term care retention. This is similar to the observations made in other SCD cohorts in Kilifi (Kenya), Kumasi (Ghana), and Nigeria [20,40,41], underscoring the importance of scaling up and sustaining SCD care services.

In our study, we observed that less than a quarter of patients had health insurance coverage. Similarly, studies in SSA have consistently reported low health insurance coverage as a major barrier to accessing care for chronic conditions like SCD. Despite the provision of healthcare services being provide for free in Zanzibar to those without health insurance, access to care may be limited when government-backed services are lacking. During these times, those with health insurance tend to have more ready access to care compared to those who have to pay out of pocket, highlighting the critical role played by health insurance in the care of patients with SCD. As the standards of care for SCD continue to improve in Zanzibar, it is important to sensitize patients to enrollment in health insurance plans to secure long-term care financing.

This study had several limitations. First, our study was based on a single-center cohort of SCD patients at a tertiary hospital in Zanzibar, which lacked representation of the true landscape of SCD across Zanzibar. Furthermore, in the absence of surveillance data based on a large geographical population, it is difficult to ascertain whether the complications observed among study participants are representative of the entire SCD population on the islands. However, our approach was necessary at the time of the study, since MMH was the only facility in Zanzibar offering diagnostic services and comprehensive care for SCD. Future studies encompassing primary and secondary healthcare settings across Zanzibar are strongly recommended to complement our findings. Such studies would provide a more comprehensive and representative understanding of the SCD population across the island, addressing the limitations of our single-center cohort and ensuring broader applicability of the results. A scale-up of newborn screening services across Zanzibar will provide an opportunity for more representative cohort studies. Second, with the exception of hemoglobin levels, most clinical outcomes evaluated in this study were based on patients’ self-reporting and hence prone to recall bias. To mitigate this, we confined ourselves to the evaluation of major clinical events, such as pain crisis, hospitalization, and blood transfusion, which were less prone to recall bias over the 4-month follow-up window. The study team also extensively validated the clinical data from the available medical records before the analysis to reduce the chance of errors. Nonetheless, we emphasize the need for future studies to incorporate objective measures, such as prospective data collection, to validate and enhance the reliability of these findings. Additionally, future research should include functional validation approaches, such as enzyme kinetics assays or structural modeling, to provide greater insights into the biological implications of the identified findings.

Lastly, the lack of multivariate analysis in this study could have hindered the determination of novel predictors of SCD clinical outcomes in Zanzibar. However, we plan to explore this in the future when the number of patients with the clinical outcome events of interest is expected to increase.

## 5. Conclusions

Most patients with SCD in Zanzibar are homozygous (Hb SS) for SCD. Thorough evaluation of SCD prevalence in Zanzibar through the implementation of newborn screening programs and improvements in care are warranted. To improve patient outcomes, efforts should be made to scale up basic SCD care services such as ensuring the availability, affordability, and utilization of hydroxyurea, penicillin prophylaxis, and PPSV-23. Additionally, targeted strategies in Zanzibar, such as capacity-building initiatives to train healthcare providers in the management of SCD, policy advocacy for the inclusion of hydroxyurea and PPSV-23 in Zanzibar’s essential medicines list, and the implementation of community-led awareness campaigns to address misconceptions about SCD and enhance treatment adherence are recommended.

## Figures and Tables

**Table 1 genes-16-00047-t001:** Sociodemographic characteristics of the study participants.

Characteristics	Number of Patients, N(%)
Age (years)	
0–5	198 (27.3)
6–13	322 (44.5)
14–17	81 (11.2)
18 and above	123 (17.0)
Gender	
Male	367 (50.7)
Female	357 (49.3)
Health Insurance ownership	
Yes	142 (20.4)
No	582 (80.4)
SCD-Phenotype	
SCD-SS	713 (98.5)
SCD-SC	9 (1.2)
SCD-S β-plus thalassemia	2 (0.3)

**Table 2 genes-16-00047-t002:** Management of patients with SCD at MMH.

Medicine	N (%)
Hydroxyurea	
Yes	191 (26.4)
No	533 (73.6)
Pneumococcal Vaccination (PCV-13)	
Yes	192 (26.5)
No	532 (73.5)
Folic Acid	
Yes	724 (100.0)
No	0 (0)
Anti-malarials	
Yes	4 (0.6)
No	720 (99.4)
Penicillin V	
Yes	188 (26.0)
No	536 (74.0)

**Table 3 genes-16-00047-t003:** Association between follow-up visits and clinical among SCD patients attending MMH.

Clinical Event	Yes (n%)	No (n%)	*p*-Value
Did you experience any pain since the last visit?			
Patients reporting Pain at 2nd visit	24 (4.3)	539 (95.7)	0.021
Patients reporting Pain at 3rd Visit	11 (1.9)	565 (98.1)	
Have you been admitted since the last visit?			
Patients admitted at 2nd visit	27(4.6)	557 (95.4)	0.042
Patients admitted at 3rd visit	32(5.5)	554 (94.5)	
Have you received blood transfusion since the last visit?			
Patients received BT during 2nd visit	9 (1.6)	551 (98.26)	0.962
Patients received BT during 3rd visit	10 (1.7)	566 (98.26)	
Severity of Anemia			
Hb level < 5 g/dL at 2nd visit	46 (7.7)	548 (92.3)	0.058
Hb level < 5 g/dL at 3rd visit	28 (5.0)	531 (5.0)	

## Data Availability

The data presented in this study is not readily available due to the reason that, the primary researcher is still analyzing data for subsequent publications. Hence, data will be available only on request from the corresponding author through email address ahlaam4@yahoo.com.

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
