# Peer review of "Sickle Cell Disease in the Islands of Zanzibar: Patients’ Characteristics, Management, and Clinical Outcomes"

_genes, 2025, doi:10.3390/genes16010047_

Round 1
Reviewer 1 Report
Comments and Suggestions for Authors
This manuscript examines the demographics, management, and clinical outcomes of Sickle Cell Disease (SCD) patients at Mnazi Mmoja Hospital (MMH) in Zanzibar, using data collected between September 2021 and December 2022. It identifies prevalent SCD phenotypes, demographic distributions, clinical care approaches, and patient health outcomes, emphasizing the underutilization of comprehensive care measures. The study offers valuable insights into SCD management in a low-resource setting, which can inform regional health policies highlighting the need for newborn screening programs and improved access to essential medications, contributing to potential health policy enhancements.
Comments
1. The study was conducted solely at Mnazi Mmoja Hospital, which may not fully represent the broader SCD population in Zanzibar. This limits the generalizability of the findings to other healthcare settings and areas on the islands.
2. The reliance on self-reported data for outcomes such as pain episodes and hospital visits could introduce recall bias, affecting the reliability of the data. This limits the accuracy of patient-reported clinical outcomes, as they may be under- or overestimated
3. While the study provides valuable statistics, it lacks more advanced statistical analyses, such as multivariate approaches, to control for potential confounding variables. This omission limits the depth of the findings and their interpretability concerning the factors influencing clinical outcomes
4. Although the manuscript acknowledges limited access to treatments like hydroxyurea and pneumococcal vaccines, it does not thoroughly explore the barriers preventing their use. This could include economic constraints, supply chain issues, or misconceptions about treatment.
5. The discussion section does not deeply compare the findings with those of similar studies in other sub-Saharan African countries or regions with similar socio-economic and healthcare challenges. This limits the manuscript’s ability to position its findings within a broader context
6. The study’s follow-up period is relatively short, spanning from September 2021 to December 2022. This timeframe may not capture long-term trends in disease management and outcomes, limiting the scope for assessing sustained healthcare improvements or long-term impacts of interventions
7. While the conclusion highlights the need for scaling up services like newborn screening and basic SCD care, it lacks detailed, actionable recommendations. More specific strategies for healthcare policymakers and practitioners could improve the applicability of the findings.
8. The manuscript touches on the lack of health insurance among many patients but does not delve into how socioeconomic factors affect access to care or adherence to treatment protocols. This additional analysis could offer a more comprehensive understanding of the challenges faced by patients.
Author Response
I have uploaded the the reviewer comments and my answers as per requirement.

Reviewer 2 Report
Comments and Suggestions for Authors
The present article describes phenotypes, sociodemographic characteristics, healthcare and clinical outcomes of SCD patients in a hospital in Zanzibar. The manuscript is well written and readable. This manuscript is unique and novel. This paper covers a wide range of important research fields such as socio-demographic characteristics, management of patients, and association between follow-up visits and clinical results. I think that this interesting article would certainly advance our understanding of the physiologically and pathologically important SCD disease from clinical perspective.
I strongly recommend this great paper to be published in the Journal. However, I raise minor concerns that need to be addressed before publication. If those concerns should be adequately addressed in the revised manuscript, this superb paper would be further significantly strengthened.
Concerns that need to be addressed are
[1] Authors’ names: Names should be straightly listed up. The present list of authors is complicated and confusing. Please remedy this.
[2] I feel curious why Appendix A is incorporated later. Those tables are already incorporated into the main text. Duplicated?
[3] I am not sure if the journal “Genes” is appropriate for publication of the present paper.
For example, please consider publication in “Healthcare”, “Hemato”, “Hematology Reports”, “Hospitals”, “Medical Sciences”, “Medicines”, or relevant journal of MDPI.
I have no reason to decline publication of this interesting and novel researches. I recommend this great paper to be published in the Journal after minor concerns are addressed.
Author Response
I have attached the document with reviewer comments and my responses as per requirement

Round 2
Author Response
Thank you for all the valuable comments, I have attached the word document with response.
